# Vehicular Crowdsourcing for Congestion Support in Smart Cities

**Stephan Olariu**

Department of Computer Science, Old Dominion University, Norfolk, VA 23529, USA; olariu@cs.odu.edu

**Abstract:** Under present-day practices, the vehicles on our roadways and city streets are mere spectators that witness traffic-related events without being able to participate in the mitigation of their effect. This paper lays the theoretical foundations of a framework for harnessing the on-board computational resources in vehicles stuck in urban congestion in order to assist transportation agencies with preventing or dissipating congestion through large-scale signal re-timing. Our framework is called *VACCS: Vehicular Crowdsourcing for Congestion Support* in Smart Cities. What makes this framework unique is that we suggest that in such situations the vehicles have the potential to cooperate with various transportation authorities to solve problems that otherwise would either take an inordinate amount of time to solve or cannot be solved for lack for adequate municipal resources. VACCS offers direct benefits to both the driving public and the Smart City. By developing timing plans that respond to current traffic conditions, overall traffic flow will improve, carbon emissions will be reduced, and economic impacts of congestion on citizens and businesses will be lessened. It is expected that drivers will be willing to donate under-utilized on-board computing resources in their vehicles to develop improved signal timing plans in return for the direct benefits of time savings and reduced fuel consumption costs. VACCS allows the Smart City to dynamically respond to traffic conditions while simultaneously reducing investments in the computational resources that would be required for traditional adaptive traffic signal control systems.

**Keywords:** IoT; Smart Cities; vehicular crowdsourcing; traffic congestion

## 1. Introduction and Motivation

Most traffic signals in the US run a set of predefined timing plans that set the signal's cycle length and green phase length based on historical traffic volumes that vary with time of the day and day of week. In most cases, the optimization of the signal systems currently occurs off-line at either the isolated intersection or corridor level [1]. One of the major disadvantages of this approach is that it requires data on traffic-turning movements be regularly collected to develop optimized traffic signal plans off-line. A second major disadvantage is that the time-of-day based signal timings do not adapt well to unexpected changes in traffic demand. For example, if an incident on the roadway network causes travel patterns to change significantly, the signals often cannot fully accommodate the changes in flow, resulting in traffic buildup and congestion. In order to ensure that the signals function as well as possible, they have to be re-timed regularly to reflect current conditions. Unfortunately, due to budget or manpower limitations, transportation agencies often neglect to re-time signals, resulting in unnecessary delays to the traveling public. As recently as 2019, the Federal Highway Administration estimated that more than 75% of the country's 330,000 traffic signals are operating with outdated or uncoordinated signal timing plans [2].

While the transportation agencies have devoted substantial effort to optimizing traffic signals at the corridor level [1,3], to the best of our knowledge, the problem of rescheduling the timing of traffic lights at the scale of a wider urban area is still very much uncharted territory. The principal reason behind this state of affairs is the combinatorial explosion inherent in the process of transiting from small-scale to large scale, complex problems.

This is clearly the case when the traffic flows from several corridors compete for unshareable resources—time and road bandwidth.

In an ideal world, the Smart City's municipal *Traffic Management Center* (TMC) would have at its disposal sufficient computational resources that could be used to compute, in near real-time, optimal timing plans for all the traffic signals under its jurisdiction. In reality, no municipality can afford the huge expenditure involved in purchasing and maintaining a huge computational resource solely dedicated to optimizing traffic flow. One alternative would be for the TMC to outsource this huge computational task to one of the existing cloud service providers. This approach, however, would not only be costly but, due to the overheads and time delays incurred, would not guarantee traffic signal re-timing in useful time, say, to mitigate the effects of non-recurrent (i.e., unexpected) congestion.

Similarly, the transportation agencies have not had the means to address non-recurrent congestion due to crashes and disruptions of traffic flow due to emergency vehicle pre-emption. While efforts in Adaptive Traffic Signal Control (ATSC) systems (e.g., SCOOT [4], SCATS [5,6], and later UTOPIA [7]) were intended to address non-recurrent traffic conditions, they have not been very successful in the US because of significant installation, maintenance, and operation costs and the lack of improvement they offer. In ATSC systems the controllers receive traffic-related data from various sensor systems [8] and adjust signal timing based on simple logic such as gap-out and max-out.

Under present-day practices, the vehicles on our roadways and city streets are mere spectators that witness traffic-related events without being able to participate in the mitigation of their effect. In recent years, there have been a number of efforts worldwide to leverage connected vehicles for the purpose of approximating traffic state parameters in near real-time [9–11]. Notable in this regard is the European effort known as CVIS [12] that relies heavily on the communication between vehicles and road-side infrastructure.

This paper takes a complementary approach: we introduce and develop the theoretical basis of a framework for harnessing the on-board computational resources (including smartphone and other IoT devices) in vehicles stuck in urban congestion in order to assist transportation agencies with dissipating congestion through large-scale signal re-timing. We call our approach *VACCS:(VACCS rimes with VAX.) Vehicular Crowdsourcing for Congestion Support in Smart Cities*. Here, *support* denotes a combination of prevention and mitigation activities. What makes our vision unique is that we suggest that in such situations the vehicles have the potential to cooperate with various transportation authorities to solve problems that otherwise would either take an inordinate amount of time to solve or cannot be solved for lack for adequate municipal resources. An obvious advantage is cost savings to municipalities and incentives to drivers.

Instead of re-timing signals at the corridor level only, VACCS offers the opportunity to optimize traffic flow at the Smart City level by making dynamic use of vehicular network probe data to re-time signals. VACCS will enable traffic signals to be more responsive to actual conditions, rather than being based on historic volume counts.

VACCS offers direct benefits to both drivers and the Smart City. By developing timing plans that respond to current traffic conditions, overall traffic flow will improve, carbon emissions will be reduced and the economic impact of congestion, in terms of wasted fuel and lost productivity hours, will be lessened. It is expected that drivers will be willing to participate in *vehicular crowdsourcing* (VCS, for short) to develop improved signal timing plans in return for the direct benefits of time savings and reduced fuel consumption costs. VACCS will allow the municipality to dynamically respond to traffic conditions while simultaneously reducing investment in the computational resources that would be required for traditional adaptive traffic signal control systems. In the end, the citizens of the Smart City reap direct personal benefits, while local governments can reduce expenditures and improve overall network flow.

*Our Contributions*

The goal of this paper is to lay the theoretical foundations of VACCS, a framework for vehicular crowdsourcing in support of congestion resolution (i.e., prevention and mitigation) in Smart Cities. Specifically, we make the following main contributions:

First, we develop techniques for setting up and managing vehicular crowdsourcing: we will use vehicles not only as computational resources but also as sensing resources as well. Through monitoring signal cycle failure or other similar technique, VACCS-enabled vehicles will participate in determining when congestion is imminent to call for signal re-timing in an effort to avoid congestion altogether, or to mitigate its effect. Once the imminent congestion has been detected, the TMC may decide to enable vehicular crowdsourcing to evaluate potential re-timing schedules. Issues involved in managing the vehicular crowdsourcing process include developing communication and protocol requirements, ensuring security and privacy for vehicles and investigating the particular approaches to be used in assessing the effects of the re-timing plans.

Our second main contribution is to develop theoretical models for dynamic resource discovery and availability prediction. We will discuss techniques for discovering the resources available on each VACCS-enabled vehicle close to a congested intersection and predicting which vehicles will be available for the duration of the computational task. We will also use a model for estimating the expected number of VACCS-enabled vehicle at an intersection. Our preliminary results show that assessing the amount of available resources, assigning them to particular applications allocating tasks to vehicles, and migrating tasks from vehicles that are departing the system are very challenging problems that can only be solved stochastically [13].

VACCS is built around the concept of IoT-supported vehicular crowdsourcing, a novel, transformative, next-generation computing paradigm that is not possible with currently available technology. In support of preventing congestion or mitigating its effects, VACCS involves tasking a pool of vehicles to perform parallel versions of complex traffic optimizations that, properly integrated, can lead to efficient signal re-timing.

We focus our attention on non-recurrent congestion due to a variety of causes including emergency vehicle preemption and railroad crossing. Giving priority to emergency vehicles, public transportation vehicles or the train may result in heavy congestion and disruption of traffic flow at the intersection, corridor, and network level. While existing traffic controllers offer mitigation strategies, the approach involves recovering bandwidth progressively without considering its effect on actual traffic congestion [13].

We trust VACCS will enhance our knowledge and understanding of vehicular crowdsourcing, bridging the gap between municipal-level needs and problems related to the distributed ownership of computational resources. In fact, we anticipate that our results could be applied to other transformative Smart City or Smart Community applications [14].

In addition, VACCS will provide insight into the coordination between tomorrow's vehicles in Smart Cities and their on-board computational, storage, and networking resources. Enabling this coordination faces many challenges, including the requirement of wireless communications and the mobility of individual vehicles which affects the dynamics of groups of vehicles. We anticipate many other applications of this technology that could have a profound and lasting societal impact.

The remainder of the paper is organized as follows: To set the stage for our discussion, in Section 2 we provide a high-level introduction to Smart Cities, the Internet of Things, edge computing, vehicular crowdsourcing, and smart mobility, one of the key services in the Smart Cities of the future. Next, Section 3 reviews technicalities that will facilitate the presentation. Specifically, in Section 3.1 we discuss basic assumption about the capabilities of vehicles; in Section 3.2 we introduce vehicular networks and the wireless communication model assumed throughout the paper; in Section 3.3 we introduce the assumed TMC model; Section 3.4 discusses the assumed capabilities of the traffic light controllers; Section 3.5 serves as a basic introduction to traffic monitoring; Section 3.6 introduces basics of traffic signal optimization. Further, Section 4 offers a high-level description of the VACCS

architecture. This is then followed, in Section 5, by a high-level description of VACCS in action. Next, Section 6 discusses the technical details of the various tasks involved in VACCS that were glossed over in the previous sections. Finally, Section 7 offers concluding remarks and identifies a number of challenges ahead.

## 2. The Rise of Smart Cities

The main goal of this section is to offer a succinct review of Smart Cities, IoTs, edge computing, and smart mobility–one of the key services offered by Smart Cities.

### 2.1. Smart Cities

The Smart City metaphor was proposed in the early 1990s as an illustration of how urban communities were turning towards technology, innovation, and globalization [15]. Smart Cities are "urban centers that use intelligent, connected devices and automated systems that maximize the allocation of resources and the efficiency of services" [16].

In the light of recent urbanization reports, United Nations' statistics predict that, by 2050, over 68% of the world population will reside in metropolitan areas [17]. Thus, the importance and relevance of Smart Cities is poised to increase dramatically in the years to come.

While Smart Cities have been defined in myriad ways, it is telling that all these definitions have two explicit or implicit characteristics in common: first, the Smart Cities assume a transparent governance and management style that *anticipates* the read needs of the citizens; and, second, they assume a broad and continued *engagement* and active *participation* of the citizens. These two characteristics of Smart Cities can be summarized as "putting the citizen first", or being *human-centric*, or *citizen-centric*. Empowering their citizens with increased access to high-quality information and services is one of the defining dimensions of a Smart City [18].

The phenomenal technological advances of the past two decades are enabling the transition from present-day urban communities to the Smart Cities predicted by the visionaries. To make the transition to Smart Cities, and to address the challenges involved in this transition, present-day urban communities must harness and put to work their most creative ideas and initiatives. Part of the challenge is to understand how advanced data and Information and Communications Technology (ICT) can be used to empower the citizens, reduce traffic congestion, protect the environment, respond to climate change, attend to the needs of under-served communities, support economic vitality, etc. [19,20].

Smart Cities are expected to provide a common societal infrastructure for prosperity based on an advanced service platform which turns out to be their main workhorse. The insight behind Smart Cities is that continued progress of ICT and digital technologies of all sorts will provide individuals and society tremendous opportunities for innovation, growth, and unprecedented prosperity. All this will be provided through various forms of human-to-human, human-to-machine, and machine-to-machine cooperation and services [21,22]. Most of these forms of cooperation and services developed between humans and machines or between autonomous machine systems have yet to be defined and understood [19,23].

Recently, a number of cities around the world have branded themselves as Smart Cities. Examples include Amsterdam (The Netherlands), Song-do (South Korea), Copenhagen (Denmark), Masdar (Abu Dhabi), Singapore (Singapore), Makkah and Medina (Saudi Arabia), Madrid and Santander (Spain), Glasgow (Scotland), London (United Kingdom), Yokohama (Japan), among numerous others. To give the reader an idea of the magnitude of this trend, suffice it to mention that in 2015 India gave the green light towards the designation, by 2020, of 100 Indian cities as Smart Cities [24].

We subscribe to the idea that, fundamentally, Smart Cities are Cyber-Physical Systems (CPS) where the deployed infrastructure represents the Physical component, while the people, their government, as well as the apps they use constitute the Cyber component. These two components are involved in a dialectic relationship where they feed, condition, and learn from each other. Admittedly, this is not a widely-held view, yet it is a very appealing one, especially because these components are (or should be) engaged in a symbiotic

relationship. In our opinion, an interesting corollary of this is that the two components thrive or fail together. A direct consequence of this perspective is that the Smart Cities will continually innovate and enhance their service offerings to the citizens based on community intelligence. This behavior will lead to the evolution of diverse Smart Cities based on their own characteristics and the needs of their citizens.

### 2.2. The Internet of Things—A Key Enabler of Smart Cities

The Internet of Things (IoT) has been defined in myriad ways [25]. In a nutshell, an IoT is a network of everyday objects endowed with computation and communication capabilities. The IoT devices can be sensor nodes, actuators, coffee makers, vacuum cleaners, refrigerators, edge devices such as RFID tags, smartphones, smart watches, tablets, smart meters and other similar devices [14].

However, the diversity and heterogeneity of devices participating in IoT through dynamic joining and leaving can have direct consequences on workload assignment, networking interfaces, privacy and security. For example, numerous researchers have pointed out that the problem of IoT system security and privacy is becoming significant in the context of Smart Cities and needs urgent solution.

IoT is generally viewed through the lens of Smart Cities [26]. Numerous researchers have expressed the view that the IoT will turn out to be the key enabling technology for the transition to Smart Cities.

We hold the view that the IoTs within a Smart City will benefit from being integrated into a Smart City-wide IoT *ecosystem* that, through the services it offers, will enable the smart services mentioned above and, in the process, will revolutionize the citizens' experience making life in the Smart City safer, more enjoyable, and more environmentally friendly [20,27]. The main challenge is the integration of IoTs into a harmonious ecosystem. While, to date, mainly stove-pipe integration strategies for IoTs have been proposed, we believe that a more efficient and effective solution for IoT integration into an ecosystem is a market-driven, open integration of IoTs based on a valuation of the goods and services provided.

### 2.3. Edge Computing—An Instance of IoT

In recent years, the realization that Moore's law no longer applies has motivated an emphasis shift in computer architecture towards energy-efficient special-purpose architectures. In conjunction with recent advances in nano-technology and smart materials, this lead, quite naturally, to the development of new types of connected smart devices, including smart watches, smart glasses, smart meters, smart robots, connected vehicles, and many other household items that deserve to be called "smart" (e.g., smart coffee makers, smart refrigerators, smart vacuum-cleaners, etc.).

These pervasive and ubiquitous smart devices are referred to, collectively, as *edge* devices, or *the edge*. We are beginning to see more and more network traffic originating at these edge devices. It was soon realized that the data generated at the edge is incredibly rich in contextual information and, hence, extremely valuable and should be harvested to capture and understand context. Unfortunately, because of the widening gap between bandwidth capacity and data volumes, the data generated at the edge will increasingly stay at the edge and will be thrown away for lack of adequate processing power.

A good example of such contextually rich data is the sensor data collected by the *vehicles* that crisscross our roadways and city streets. These data are highly ephemeral as it reflects instantaneous traffic conditions that are apt to change fast. Due to latency, costs, and the risks involved in moving data to and from a cloud, cloud-based real-time processing of edge data is neither technologically feasible nor economically viable. Indeed, the time delay incurred in offloading the task of processing the data generated at the edge to some remote cloud for processing is, simply, unacceptably long. Given the transient nature of context and context-sensitive needs of individuals and, more broadly, the community, the highest value from edge data can be extracted only by processing it in near real-time. In the light of our previous discussion, there is a critical need for an alternative computing

platform, one that allows harvesting and aggregating the huge amounts of data generated by edge devices right there—at the edge [28].

It is an interesting observation that the same edge devices that generate huge amounts of data also offer, potentially, a huge compute and storage resource that at the moment is untapped. Indeed, it is expected that the collective computing and storage capacity of smartphones will soon exceed that of worldwide servers. In Smart Cities and Smart Communities where the smart devices generate huge amounts of contextually rich data, harvesting this source of information in order to enhance the citizens' quality of experience will have very important ramifications. As another example, the large number of vehicles in our driveways, parking lots, and city streets can be seen as important edge devices with significant compute power. Their on-board capabilities can be harvested and put to good use in Smart Cities and Smart Communities.

### 2.4. Vehicular Crowdsourcing

Crowdsourcing is a process that involves outsourcing tasks to a group of people. The main difference between ordinary outsourcing and crowdsourcing is that in the former the problem to solve is outsourced to a specific body of people, such as paid employees, while in the latter the task is outsources to an undefined public.

As is typical of all emerging research areas, crowdsourcing has appeared in the literature under various other names including, peer production, community systems or collaborative systems. It has been argued by several authors that crowdsourcing can be legitimately looked at as a collaborative way of problem solving [29].

*Vehicular Crowdsourcing* (VCS) is an instance of a crowdsourcing where a group of vehicles lend their on-board processing resources to an authorized user. While in this paper we limit authorized users to the Smart City's TMC for the express purpose of preventing or mitigating congestion through signal re-timing, the concept is clearly more general. Using the vehicles themselves as computation resources in the context of crowdsourcing opens a new area of research. The principal goal of VCS in support of detecting imminent congestion or mitigating its effects, is to perform parallel and distributed versions of complex optimizations that can lead to an efficient re-timing of traffic signals.

A related, but quite distinct, area is that of *crowd computing* which combines mobile devices and social interactions to achieve large-scale distributed computation [30]. Here, an opportunistic networks of mobile devices, including smart phones, PDAs, laptops and other IoT devices, offers an aggregate compute power and communication bandwidth. Murray's seminal paper [30] points out a number of reasons crowd computing is attractive: key among them is the willingness of people to voluntarily contribute to a common cause, even if no reward is offered or expected. Typically crowd computing involves one or a series of tasks that are farmed out to a number of mobile devices. As these devices socially meet other such devices, the tasks are shared with the new devices and this process continues until the tasks are completed.

### 2.5. Smart Mobility—A Key Service in Smart Cities

The rise and adoption of Smart Cities creates opportunities for creative and efficient management of the available or planned municipal resources and services. Among these urban resources, urban mobility [31] will remain one of the main challenges in Smart Cities. Smart mobility involves the use of state of the art Intelligent Transportation Systems (ITS) to make traffic more fluid, better adapted to the needs of the citizens while, at the same time, minimizing its environmental footprint.

Smart mobility has two main components, namely *smart transportation* and *smart parking*. In this paper, we are devoting our attention to smart transportation solutions. Our goal is to show how vehicular crowdsourcing can provide innovative solutions to the rescheduling of traffic lights in the wake of non-recurrent congestion.

## 3. Technical Background

In this section, we present technical background on vehicular capabilities, traffic signal optimization, vehicular communications, and using vehicular networks for distributed computation.

### 3.1. The VACCS-Enabled Vehicle Model

In its 2006 ruling [32], the National Highway Traffic Safety Administration (NHTSA) has mandated that starting in September 2010 an *Event Data Recorder* (EDR) be installed in vehicles with an unloaded weight of less than 5000 lbs. The EDR is responsible for recording mobility attributes including acceleration, deceleration, lane changes and the like. Each such transaction is associated with an instantaneous GPS reading. All of the vehicle's sub-assemblies feed their readings into the EDR [33,34]. In addition to a tamper-free EDR, it is expected that vehicle-to-vehicle (V2V) and vehicle-to-infrastructure (V2I) technology penetration will increase substantially in the years to come.

We expect VACCS-enabled vehicles to contain an on-board computer, a GPS device and a digital map, a radio transceiver based on DSRC [35] and IEEE 802.11p/WAVE [36,37] and a variety of sophisticated sensing devices, ranging from on-board radar to cameras, that can alert the driver to mechanical malfunctions and hazardous road conditions. While today such features are only present in high-end vehicular offerings, we expect that in a few years they will be commonplace. We assume that the on-board computer is of laptop class with a quad-core 2.8 GHz processor, 16 GB RAM and 1.0 TB of storage.

As a direct consequence of its on-board capabilities, each VACCS-enabled vehicle is aware of its geographic location and, at a finer granularity, of the traffic lane in which it currently resides. As discussed at the end of this subsection, VACCS-enabled vehicles have the capability to data-mine driver behavior.

We further assume that VACCS-enabled vehicles have local versions of the optimization/simulation code that they will run. Such code could be loaded as part of the yearly technical inspection or else at vehicle registration time. It is important to note that the code need not be active. It could be activated every time the vehicle actually participates in vehicular crowdsourcing. To activate the code all that is needed is a per session key that the vehicle will receive at participation time.

Data-Mining Time-Dependent Trip Parameters

As it turns out, VACCS-enabled vehicles can use their on-board EDR, on-board unit (OBU), or smart phone equipped with GPS navigation (with opt-in for sharing information in return for receiving better traffic information or route guidance) to data-mine *origin-destination* (OD) information and driver behavior for each user profile they store. For a given user profile, the EDR may store in addition to OD information, valuable information about the trip. For example, for each intersection along the way, the EDR can record statistical information about the probability of various possible transactions at a given intersection: traversing the intersection, turning left or turning right. This statistical information can be easily acquired and consolidated over a period of time. An immediate benefit of having this sort of statistical information available is that the vehicle can predict the most likely route to be taken by the current driver as a function of the time of the day, day of the week, etc.

### 3.2. Wireless Communications and Vehicular Networking

The promise of vehicular networks has captivated the networking research community over the past two decades. Applications and protocols have been proposed, standards have been written, and hardware was developed to support this vision. The societal benefits and potential for rapid commercialization will drive the production of vehicles equipped with advanced processing, communication, storage, and sensing capabilities. With almost seven million new cars purchased in the US each year [38], we will soon have millions of these advanced vehicles on our roadways.

The past decade has seen a rapid convergence of ITS with vehicular networking technology, promising to revolutionize the way we drive by creating a safe, secure, and robust

ubiquitous computing environment that will eventually pervade our highways and city streets.

Even though wireless communications are presently in a state of flux, we proceed with the assumption that the communication needs of VACCS will be supported by the Dedicated Short Range Communications (DSRC) [35] and IEEE 802.11p/WAVE [36,37] standards or some other similar wireless technology, including WiFi.

The MAC layer of IEEE 802.11p is a modified version of IEEE 802.11a (WiFi) which uses Channel Sensing Multiple Access (CSMA) for sharing access to the wireless medium. At the physical layer, DSRC is divided into seven channels, each 10 MHz wide. The control channel (CCH) is used for beacon messages, event-driven emergency messages and service advertisements. The remaining six service channels (SCH) support non-safety applications provided by roadside units (RSU), such as our traffic signal controllers. Channel switching is employed to support multiple channels. There is a synchronization interval that consists of a CCH interval (CCHI) and a SCH interval (SCHI). During the CCHI all radios must be tuned to the CCH to broadcast updates and listen for advertisements from neighbors and RSUs. During the SCHI the vehicles may tune to the SCH of their choice depending on the services offered. As currently envisioned, WAVE allows for communications of safety and non-safety applications through a single DSRC radio. Unfortunately, it has been shown that DSRC cannot support both safety and non-safety applications with high reliability at high traffic densities: either safety or non-safety applications must be compromised.

*3.3. The TMC Model*

By monitoring legacy devices including cameras, ILDs, infrared sensors the TMC may acquire approximate knowledge about the traffic flow on its major roadways. We assume that, in most cases, the Smart City does not have appropriate computational resources dedicated to the express use of engineering the most efficient way to decongest an area in the event of non-recurrent congestion. We assume, however, that the Smart City has the appropriate code to run and the authority to do so as required.

The vision that we propose in this paper is that the computational power required to run the traffic light optimization code is available, collectively, in the vehicles stuck in congested traffic. Thus, in the case of a traffic accident that is blocking one of its roadways, the TMC can authorize a suitable group of traffic lights in the vicinity of the congestion to set up and manage, on its behalf, vehicular crowdsourcing involving the vehicles stuck in traffic.

*3.4. The Traffic Light Model*

We assume that the Smart City has instrumented the urban area under its jurisdiction by deploying high-bandwidth access to individual traffic lights and supporting infrastructure. This could be accomplished by adding 3G/4G modems at the intersection if high speed WiFi/DSRC is not available. Each traffic light controller contains a DSRC-compliant or WiFi radio transceiver for communicating with neighboring vehicles, an LTE or 4G cellular broadband connection to communicate with the TMC, and a dedicated computing device such as the versatile low-power PC/104 systems [39].

Using DSRC or WiFi, each traffic light transmits periodically identification information to alert approaching vehicles. By monitoring legacy devices including cameras, ILDs, infrared sensors the traffic light controllers can acquire fairly accurate knowledge about the traffic flow in their vicinity.

In VACCS, one of the key roles played by traffic lights is to collect and aggregate traffic-related data (1) from passing vehicles, and (2) by exchanging information, on an intermittent basis, with adjacent traffic lights and with the TMC. When individual traffic light controllers become aware of an imminent *trend* or a traffic incident, they disseminate this information to the TMC. In turn, the TMC will make the determination of whether or not to mandate vehicular crowdsourcing. We also envision that a VACCS-enabled vehicle could be responsible for forming a crowdsourcing group and for coordinating with TMC so that decisions are made locally.

### 3.5. Traffic Monitoring

ITS provides traffic monitoring and data collection functions using legacy technologies such as ILDs, magnetometers, video-image processing, and probe vehicles. It is well documented [40,41] that the equipment installed in support of collecting such data is expensive and costly to maintain and repair. Not surprisingly, the USDOT is investigating a number of possible alternatives. In the next decade, the USDOT plans to develop an architecture for vehicle infrastructure integration that will collect data from passing vehicles and, after aggregation at a *central message switch*, will be distributed to the traveling public [42,43]. The architecture document states that all messages will be digitally signed, with a central certificate authority responsible for distributing public and private encryption keys. However, the proposed ITS approaches that supplement ILDs and video cameras by the use of wireless communications have raised security and privacy concerns, very similar to those encountered in vehicular networks.

Recently, cell phone technology was proposed for traffic monitoring. The idea is to track cellular handoffs between cell towers using handoff information already collected by the cell phone companies and to use elapsed times between handoffs to determine the speed of vehicles. This technology is much like that used for probe vehicles, but there is no special equipment needed in the vehicle, besides a cell phone. Another idea that exploits the prevalence of smartphones is to supplement legacy traffic monitoring with traffic incident reports submitted by the driving public. A recent implementation of this idea has lead to 511 Traffic that offers an at-a-glance view of road conditions in a given geographic area [44]. Unfortunately, 511 Traffic is a centralized system that accumulates and aggregates traffic-related feeds at the TMC level and, due to inherent delays, often displays stale traffic information [45].

Finally, the Mobile Millennium project at UC Berkeley exploits information collected by probe vehicles to infer information about traffic. Relying solely on traffic data collected by probe vehicles seems to work best in environments that experience high concentration of vehicles and less well where there may be no "critical mass" of probe vehicles.

### 3.6. Traffic Signal Optimization

Traffic signals are assigning the right-of-way at intersections, but they can be a source of significant delays if cycle and phase lengths are not suitable for current traffic conditions. Under current practice, the process of developing optimal signal timing plans is resource-intensive. First, technicians must go into the field to manually count turning volumes at intersections. Then, these volumes are input into commercial signal optimization software packages such as Synchro or CORSIM [46,47] to develop optimal timing plans offline. Various software packages use different objective functions to develop optimal timing plans for isolated intersection or corridors. As an example, Synchro can develop a timing plan based on a performance index that examines stops, delay, and 90-th percentile queue lengths [46]. Recently, Dion et al. [1] have investigated traffic simulation model use in the development of Corridor System Management Plans (CSMPs). The current state-of-the-practice is to develop separate timing plans for different times of day and days of week based on when flow profiles change. These timing plans are then uploaded into the traffic signal controllers to operate the signal [45,48].

As mentioned in the introduction, the prevalent form of traffic signal control in the US is the actuated traffic signal. These traffic signals adjust green phase between preset minima and maxima based on arriving traffic volumes. A key issue is that they do not self-optimize and rely on stored timing plans. This means that the signal will perform progressively worse if traffic volumes change over time from what was used to develop the timing plan. Furthermore, the signal will not be able to dynamically respond to changes in volume due to incidents or special events. In order to ensure that the signal functions as well as possible, the signals have to be re-timed regularly to reflect evolving traffic conditions.

One solution that has been proposed is the use of adaptive traffic signal control (ATSC). ATSC systems use extensive detection to dynamically optimize flow along a corridor. These

systems often use no fixed cycle of phase lengths and re-time signals continually based on observed traffic flow [49]. There is often a local optimization that occurs to minimize delays at an individual intersection and a secondary global optimization that occurs along a series of signals on an arterial. While some ATSC systems have been in existence for over two decades, they have not been adopted on a wide scale. More cost-effective systems have been developed with some impressive results on corridor-level deployments [47,50]. However, due to their complexity, they have not been deployed beyond the corridor level [51].

*3.7. Distributed Computation in Vehicular Networks*

VACCS assumes distributed computation over wireless vehicular networks, just as distributed computation is performed over conventional cloud computing systems. Previous work in this area includes the VGrid project [52–55]. VGrid has been used to compute efficient lane merging schedules [52], provide accident alerts [53], and identify obstacles on the roadway [54,55]. Although the authors of VGrid discuss distributed computation, the work is not in the context of vehicular crowdsourcing and does not consider vehicles running arbitrary user code, as in the vehicular crowdsourcing context.

To run real-time dynamic traffic simulations over vehicular crowdsourcing, especially during times of traffic congestion or evacuation, VACCS must rely on distributed simulations that run on VACCS-enabled vehicles. There has been some previous work [56–58] in creating distributed versions of simulations using the VISSIM traffic simulator. Preliminary work [56] demonstrated the feasibility of the general approach, but did not deploy the system over wireless nodes. Later work [57,58] looked at how to improve the accuracy of the simulation by including real-time data from existing traffic sensors. Our approach is novel since we use data collected by VACCS-enabled vehicles to feed into the distributed traffic simulations. This allows VACCS-enabled vehicles to act both as computational and as sensing resources.

## 4. The VACCS Architecture

If the vehicular crowdsourcing concept is to achieve its full potential, the basic architecture of VACCS must be engineered to offer a seamless integration of the resources of the participating vehicles. In particular, the architecture must dynamically adapt its managed vehicular resources allocated to the application according to dynamically changing requirements and systems conditions. The goal of this section we offer a high-level overview of the VACCS architecture and challenges involved in engineering vehicular crowdsourcing.

The architecture of VACCS consists of a TMC-side API translator, a VACCS manager, a variable number of vehicular crowdsourcing managers and cluster controllers as illustrated in Figure 1. VACCS will support the needs of a single client, the TMC, and focuses on running one, or a small number of related applications on a large number of vehicles, each running an instance of some optimization code (e.g., a microscopic traffic simulator) over a traffic network. As illustrated in Figure 1, the VACCS architecture comprises the following components:

- The *VACCS Manager* located at the TMC acts as an interface with the TMC. The VACCS Manager accepts requests from the TMC for more instances of crowdsourcing in a given AoI and spawns a Vehicular Crowdsourcing Manager to handle each such request;
- A number of *Vehicular Crowdsourcing Managers*, (*VCS Managers*), each of which is in charge of managing a new instance of crowdsourcing in an AoI specified by the VACCS manager. This involves discovering and managing the dynamically-changing vehicular resources in the AoI specified by the VACCS manager. For example, the VCS Managers are responsible for discovering new resources as they become available, managing available resources, predicting the availability of resources using a suitable stochastic model, migrating tasks from departing vehicles to available ones, and balancing the load on the existing clusters. Each VCS Manager, in coordination with the cluster controllers, performs the following tasks:
  - discovers new resources as soon as they become available;
  - keeps track of currently-available resources;

- – predicts the availability of resources using a suitable stochastic model;
- – migrates tasks from a departing vehicle to an available vehicle;
- – balances the loads of the existing clusters.

- A number of *cluster controllers*, co-located with individual traffic lights, each in charge of a resource cluster. Each participating intersection has a cluster controller incorporated in the traffic signal controller. Each cluster controller interfaces with the vehicles in its cluster through a *smart access point* (SAP), responsible for managing the wireless connection to the vehicles in the cluster. The SAP is in charge of radio communication with the physical resources in the cluster;
- Several *clusters*, each consisting of the VACCS-enabled vehicles within communication range of a traffic light controller. The *cluster resources* are the pooled resources of the individual VACCS-enabled vehicles in the cluster.

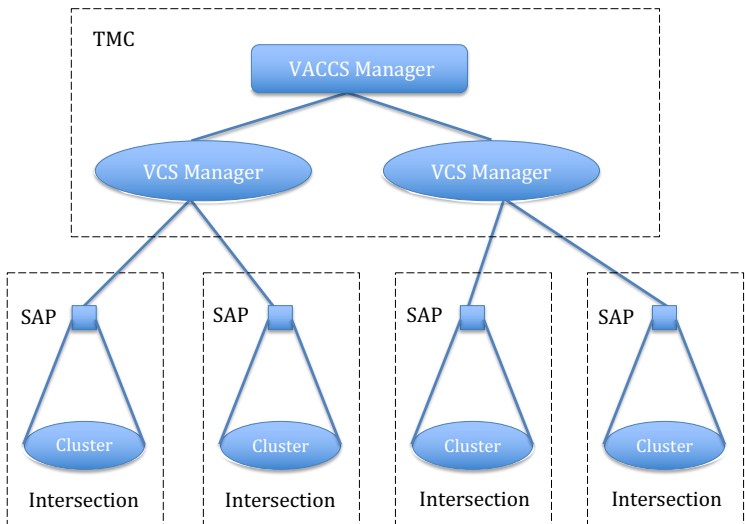

**Figure 1.** A high-level view of the VACCS architecture.

The mobility attribute of the vehicles involved in vehicular crowdsourcing, combined with the fact that the presence of VACCS-enabled vehicles in close proximity to an event is very often an un-planned process implies that the pooling of the resources of those vehicles in support of mitigating the event must occur spontaneously by the common recognition of a need for which there are no pre-assigned or dedicated resources available. This agility of action turns out to be an important defining characteristic of VACCS.

## 5. Putting VACCS to Work—A High-Level Description

The main goal of this section is to give the reader a high-level description of how VACCS works. For this purpose, in Section 5.1 we begin by presenting a hypothetical working scenario involving an urban area prone to congestion. Next, in Section 5.2 we show how VACCS operates. The technical details will be discussed in Section 6.

### 5.1. Working Scenario

To illustrate our technical solutions, we have selected an area of roughly 3500 × 4000 feet in downtown Manhattan, bounded by 58th and 72th Streets and between 5th Ave and York Ave, as illustrated in Figure 2. Imagine that while traffic is still flowing freely in adjacent streets, there are signs of traffic build-up on Lexington. The standard discernible symptom of *imminent* congestion is that the green phases of various traffic lights on Lexington do not seem sufficient to allow the queue of stopped vehicles to clear. By monitoring legacy devices including cameras, ILDs, infrared sensors and by using information collected from VACCS-enabled vehicles, individual traffic controllers collocated with traffic lights (These will be referred to, simply, as *traffic lights*.) can easily determine if there is cycle

failure. If this circumstance becomes a persistent event, the TMC is alerted and, if deemed necessary, vehicular crowdsourcing is mandated.

Vehicular crowdsourcing is predicated on there being a sufficient number of vehicles that are likely to continue to travel on the congested segments (e.g., southbound on Lexington or eastbound on E. 62-th Street) and that, therefore, will be within the radio coverage of the set of traffic lights on these congested streets. Once the determination has been made to mandate vehicular crowdsourcing, a *Call for Participation* (CfP) will be issued by some of the traffic lights (at intersections *A* and *B* in Figure 2), requesting VACCS-enabled vehicles to participate. To qualify for participation in vehicular crowdsourcing, a VACCS-enabled vehicle must predict being within coverage for at least *T* time units. Notice that, as discussed in Section 3.1, since the vehicles have a fairly good idea of their most probable course, recruiting a set of participating vehicles is feasible. The exact dynamics of how the vehicles are recruited will be discussed in Section 6.4. In fact, for reasons of fault-tolerance, the number of participating vehicles recruited will exceed the actual needs of VACCS.

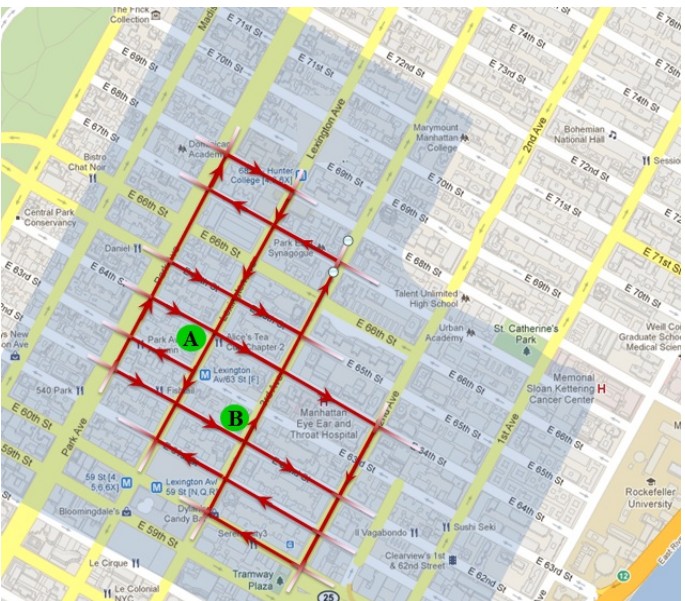

**Figure 2.** Illustrating a typical urban area prone to congestion.

Once the participating vehicles have been selected, they will be tasked with executing an *optimization code* on input provided by the TMC. The optimization code could be manifold: it could range from instances of linear programming, to heuristics for phase assignment, to microscopic traffic simulators VISSIM [59] or NGSIM [60]. At the risk of over-simplifying, we assume that all vehicles will run instances of VISSIM. It is important to note that while the code is standard, the input on with the code is run is dependent on the actual situation at hand. This input contains, for each intersection in the *Area of Interest* (AoI), defined as the blue-colored area in Figure 2, the number of vehicles traversing the intersection, turning movement counts, and signal timing parameters. The simulation performed by an individual vehicle will evaluate how well a particular signal timing plan clears the congestion. At the end of the execution (i.e., at the end of *T* time units), each participating vehicle will upload their results to the nearest traffic light. The TMC will then aggregate the solutions and will proceed to implementing the best re-timing schedule for the AoI.

We note that as the new timing plans take effect, congestion may move from one area to another. We expect that the process of detecting congestion, setting up VACCS, simulating new timing plans and re-adjusting the signal timings will be repeated until congestion has dissipated. The technical details will be discussed in Section 6.5.

*5.2. VACCS in Action—A High-Level View*

Referring to our working scenario, once imminent congestion is detected by a cluster controller (using a procedure to be described in Section 6.1), and the TMC has been alerted, the TMC identifies the boundaries of the AoI where congestion needs to be prevented or dissipated. In support of this, the TMC issues a request for vehicular crowdsourcing in the AoI to the VACCS Manager. In turn, having been tasked, the VACCS Manager spawns a new VCS Manager in charge of the AoI. The VCS Manager alerts the cluster controllers in the AoI.

As we saw, once a cluster controller has been tasked with setting up an instance of vehicular crowdsourcing, it periodically broadcasts a CfP message to recruit VACCS-enabled vehicles. Once an VACCS-enabled vehicle receives a CfP, and is willing to participate, registers (authenticates) itself with the corresponding cluster controller and transmits its traffic-related data (e.g., speed, travel times and delays, number of stops, destination, etc.) to the cluster controller for later use. Cluster controllers are responsible for collecting, aggregating and forwarding the received data to the VCS Manager which, in turn, passes it to the VACCS Manager.

The TMC, with the help of the relayed information by the VACCS Manager, evaluates the traffic conditions and identifies the number $W$ of, say, simulation instances that need to be run and the corresponding input parameters. Details will be discussed in Section 6.4. This set of parameters is then passed on to a VCS Manager which decides, in consultation with the corresponding cluster controllers, whether or not the currently available number of VACCS-enabled vehicles suffices to meet the demand for simulation instances. This decision is based on identifying *qualified* vehicles to run the simulations.

A VACCS-enabled vehicle is said to be qualified for participation in vehicular crowdsourcing if it is predicted to be within communication range of a cluster controller in the AoI for at least $T$ time units. For reasons of fault-tolerance, the goal of the cluster controllers in the AoI is to recruit at least $W$ qualified vehicles. The details of how the vehicles are recruited will be discussed in detail in Section 6.4.

Once participating vehicles near each intersection have been identified, the various inputs to be used in the simulations are communicated to the cluster controllers which will then disseminate them to individual vehicles. We assume a *Certification Authority* (CA) will be responsible for allowing qualifying VACCS-enabled vehicles to participate in crowdsourcing.

At the end of $T$ time units of executions time, the participating vehicles will upload their solutions to the nearest cluster controller. These solutions will be send to the TMC that will select the best set of timing signals for the AoI.

Though congestion may begin in a small area, in time it may spill over and affect side streets. Once the signal timings have been changed, the vehicles in the area of highest concentration of congestion will begin to move to other regions of the roadway network. The flow of vehicles leaving the congested area may well generate secondary congestion events over a larger area. If new congestion events result from the original re-timing, the TMC will restart the process of setting up vehicular crowdsourcing local to each congested area. Thus, the wide-area resolution of congestion can be perceived as a multi-stage process, with each of the successive stages addressing a less acute instance of congestion than the previous one. Therefore, one severe instance of congestion is reduced, in stages, to several less acute instances, spread over a much larger area. The framework that we have established in VACCS lends itself well to this recursive process. The details will be discussed in Section 6.5.

## 6. VACCS—The Technical Details

The main goal of this section is to discuss the details of setting up and managing VACCS, that were just hinted at in Section 5. One word of caution is in order: we recognize that there are many challenges involved in making our VACCS framework reality. For example, security and privacy are two major concerns when allowing multiple users to cooperate, and VACCS is no exception. However, in this paper we are focusing on

architectural and communication challenges only. Some of the challenges of providing privacy and security in vehicular crowdsourcing have been addressed elsewhere [61].

We now discuss how VACCS will set up and manage vehicular crowdsourcing to prevent congestion or to mitigate its effect. Since most of our roadways and city streets operate close to capacity [45], congestion is triggered by change variations in traffic flow, such as caused by an incident. We refer to this as *non-recurrent congestion* (or, simply, congestion) and distinguish it from rush-hour congestion for which, presumably, the traffic signals have been optimized. The major tasks here involve detecting imminent congestion, establishing the communications involved in setting up vehicular crowdsourcing and investigating computational models for traffic signal optimization.

### 6.1. Detecting Imminent Congestion

One of the important decisions is when to alert the TMC that vehicular crowdsourcing might be useful to address imminent congestion. During rush hour, the main problem is that there is more demand than the maximum capacity of the roadway can serve. Non-recurrent congestion, such as during a traffic incident, is often caused by a decrease in the capacity of the road, often through a lane closure. Once vehicles are past the bottleneck area, the congestion dissipates. Our goal is to time the signals such that vehicles may quickly pass by the bottleneck area or take an alternate route to avoid it altogether.

The 6-th edition of the Highway Capacity Manual (HCM-2016) [62] defines various levels of congestion in terms of the *Volume to Capacity* ratio (VtC, for short). Specifically, a roadway or street where the VtC is between 0.85 and 0.95 is considered to be "near capacity", while a value of VtC in the range 0.95 to 1 indicates that the roadway is at capacity. Of course, a VtC larger than 1 indicates over-capacity where we expect congestion to manifest itself in various degrees. To the best of our knowledge, under present-day practices, the TMC does not proactively monitor the VtC and, as a result, congestion is often detected when it is too late, precluding meaningful congestion-preventive actions. One such preventive action could be to start re-timing the traffic lights when the VtC reaches a critical level. One of the contributions of this paper is to address the issue of taking preventive action, well ahead of experiencing stopped traffic.

### 6.2. Proactively Monitoring the VtC

With legacy data collection resources it is difficult or even infeasible to proactively monitor the local VtC along each city block. We now propose a possible way of achieving this goal under a vehicle-centric data collection regimen. Specifically, every traffic light actively seeks input from VACCS-enabled vehicles. Each such vehicle, when approaching an intersection, will drop off with the local infrastructure an ordered pair consisting of its current speed and the time since passing by the previous traffic light. We expect these ordered pairs to be highly correlated allowing for trimming out outliers originating at malicious vehicles or, perhaps, at vehicles whose equipment is decalibrated. Now, factoring in the penetration rate, each traffic light computes an approximation of the current traffic volume and the average speed of the passing vehicles. Once this is done, it is a simple matter to estimate the current VtC.

By using simple data aggregation, the traffic lights keeps track of the most recent values of the VtC as well as the *gradient* of the change. The reason for the latter is that sudden sharp increases in the VtC (even at low initial values) may indicate rapid traffic buildup that, if left unchecked, may result in congestion.

The TMC may use its own previously deployed sensors (ILDs, cameras, police reports, etc.) to detect the VtC at a particular intersection, but as VACCS-enabled vehicles can communicate with the cluster controllers, they can provide additional real-time information about the conditions at a particular intersection. Average speed is a difficult metric to use on signalized roads, such as in our working scenario, because vehicles may stop or reduce their speed at intersections due to signal timings, without any congestion being experienced. To overcome this limitation, Hall and Vyas [43] suggested that a "congestion

alarm" be set whenever a vehicle must wait more than one cycle before being able to pass through an intersection. This condition is called a signal cycle failure and indicates that not all vehicles queued at an intersection during a red phase were able to depart during the next green phase.

Zheng et al. [63] developed an algorithm for detecting signal cycle failures using video image processing. They advised keeping track of the end of the queue during a red phase. If the last vehicle does not clear the intersection during the green phase, then there is a cycle failure.

We suggest using an approach similar to the one in [63], with the difference that the VACCS-enabled vehicles themselves report signal cycle failures instead of relying on infrastructure to detect this. There are several ways to achieve this goal. First, we can assume that the range of communications of a cluster controller will cover all of the vehicles between it and the previous signal. In this case, the cluster controller can broadcast the beginning of the signal phases (red, yellow, green). The vehicle at the end of the queue will hear the red broadcast. If it has not passed through the intersection before hearing the next red broadcast, it will alert the cluster controller that there has been a cycle failure. The problem becomes more challenging if we assume that the communications range of a cluster controller will not reach all vehicles in its block. In this case, we may have to rely on the vehicle at the end of the queue communicating with the previous cluster controller or performing some heuristics to determine when it has been in the block for the duration of an entire cycle. Random fluctuations in arrival volumes may cause isolated cycle failures (for example, if emergency vehicle preemption shortens a green phase), but the queue may be cleared on subsequent cycles. This is not a situation where a vehicular crowdsourcing needs to play a role. Because of these occurrences, it may be useful to set a threshold for repeated cycle failures before we indicate to the TMC that signal optimization may be needed.

### 6.3. Estimating the Number of VACCS-Enabled Vehicles in the AoI

We anticipate that, as a result of proactively monitoring the VtC at intersection, the TMC can have an accurate synopsis of the VtC values in the area monitored. By using a thresholding mechanism in conjunction with statistical data collected over time, the TMC will make the determination of setting up a vehicular crowdsourcing in anticipation of a major traffic buildup that necessitates signal re-timing. Referring back to our working scenario, the TMC having received an indication of persistently high VtC values along Lexington, has decided that the situation at hand warrants setting up an instance of vehicular crowdsourcing.

Once the determination of setting up an instance of vehicular crowdsourcing has been made, the first natural step is *resource discovery*. Notice that the high VtC readings along Lexington guarantee that, with high probability, a sufficiently large number of VACCS-enabled vehicles must be present. Thus, resource discovery is implicit in our proactive scheme.

Of course, the challenge is to obtain an accurate estimate of the number of VACCS-enabled vehicles in the AoI. There are several possibilities to do this. One possibility would be to use a combination of legacy measurements (e.g., length of queue information or surveillance camera images) along with along with statistical information about the penetration rate of VACCS-enabled vehicles.

Alternatively, we can use an analytical approach as we discuss next. Assume that vehicles arrive at the AoI according to a random process and reside in the AoI for an unspecified amount of time. We provide an analytical derivation of the probability that there are $k$ VACCS-enabled cars in the AoI at time $t$. For simplicity, we assume that at time $t = 0$ there are no VACCS-enabled vehicles in the AoI.

We begin by looking at a *baseline* scenario. Assume that vehicles arrive at the AoI in accordance with a Poisson process with parameter $\lambda > 0$. As vehicles enter the AoI, a random process "marks" some of them at random, independently of all other parameters of the underlying stochastic process and also of the residency time of the vehicle. The intention is for the marked vehicles to be precisely the VACCS-enabled vehicles. The marking

process above is such that a vehicle that enters the AoI at time $\tau$, $(0 < \tau < t)$, is marked with the time-dependent probability $p(\tau)$. Notice that the random marking ensures that the number of VACCS-enabled vehicles in the AoI is time-dependent; as vehicles arrive at the AoI, a second random process assigns residency times to them. We assume a general residency time distribution. In other words, the time spent in the AoI by a generic vehicle is a random variable $G$ with distribution function $F_G$ and finite expectation. These residency times are independent of the residency times of other vehicles and also of the arrival rate and the marking process discussed above.

Let $\{X(t) \mid t \geq 0\}$ be the counting process that keeps track of the number of marked cars resident in AoI at time $t$. We are interested in the probability $P_k(t) = \Pr[\{X(t) = k\}]$ that there are $k$ marked cars in the AoI at time $t$ and in the expected number $E[X(t)]$ of marked cars resident in the AoI at time $t$. As it turns out, $\{X(t) \mid t \geq 0\}$ is itself a Poisson process and one can prove (see Arif et al. [13]) that $\{X(t) \mid t \geq 0\}$ is a non-homogeneous Poisson process with parameter

$$\Lambda(t) = \lambda \int_0^t p(t - u)[1 - F_G(u)]\,\mathrm{d}u.$$

We note that the assumption about there being no cars in the AoI at time $t = 0$ can be removed by integrating over $(-\infty, t)$ instead of $(0, t)$ and by defining $p(\tau)$ for negative $\tau$ as well.

Taking into Account the Effect of Traffic Lights

The flow of urban traffic is regulated by traffic lights intended to optimize the use of the existing infrastructure. However, one of the unintended side-effect of traffic lights is that cars are clustered and the traffic becomes fairly bursty. Thus, in most traffic regimes, the arrival of cars in the AoI will take the form of clusters of arriving cars instead of single cars. This, of course, invalidates the assumption about the Poisson arrival employed in the baseline scenario discussed above.

We now show how to extend the results of the baseline scenario presented above to the case of clustered arrivals. Our plan is to reduce the problem of group arrival to several instances of single-car arrivals to which the results of the basic scenario apply. We make the following assumptions that underlie our analytical derivations:

- We assume that the clusters, but not individual cars, arrive at the times of a Poisson process with parameter $\lambda$. This assumption is justified on two grounds: first, assuming that the traffic lights are optimized, the timing of the lights is random depending on the instantaneous traffic intensity and, second, we assume that several streets feed into the AoI, each with its own traffic light. Under these conditions, the aggregated clustered traffic into the AoI is sufficiently random to warrant approximating the combined traffic by a Poisson process;
- At time $t = 0$ there are no VACCS-enabled cars in the AoI. As noted before, this is done for convenience only;
- As before, as cars enter the AoI, a random process assigns residency times to them. The time spent in the AoI by a generic car is a random variable $G$ with distribution function $F_G$ and bounded expectation;
- As individual cars are entering the AoI, a random process marks some of them independent of all other parameters of the underlying stochastic process and also of the residency time of the car;
- The marking process above is such that a car that enters the AoI at time $\tau$, $(0 < \tau < t)$, is marked with the time-dependent probability $p(\tau)$.

Let $C_1, C_2, \cdots, C_{N(t)}$ be the clusters of cars that have entered the AoI up to time $t$ and for all $k$, $(1 \leq k \leq N(t))$, let $Y_k$ be the number of cars in cluster $C_k$. We assume

that $Y_1, Y_2, \cdots, Y_{N(t)}$ have a common distribution $Y$. In particular, we assume that the probability that a generic cluster, $C$, contains $k$ cars is

$$\Pr[\{Y = k\}] = c_k, \tag{1}$$

independent of the time of arrival of the cluster, of the arrival rate of the clusters and of the number of cars in other clusters. Let $S_1, S_2, \cdots, S_{N(t)}$ be the number of marked cars in each cluster that are still resident in the AoI at time $t$.

It can be shown that the random variables $S_1, S_2, \cdots, S_{N(t)}$ are independent and identically distributed. Let $\{R(t) \mid t \geq 0\}$ be the counting process that keeps track of the number of marked cars resident in the AoI at time $t$. To compute the expectation $E[R(t)]$ of $R(t)$ we decompose the problem at hand into a sequence $B_1, B_2, \cdots$ of instances of the baseline scenario, each such instance $B_i$ being obtained by picking one car from each arriving cluster. Even though the clusters do not contain the same number of cars, the resulting single-car instances are Poisson streams (as copies of the original Poisson stream or as random sub-streams thereof). In either case, it is clear that each of the $B_i$s is a Poisson stream and, as a result, the individual car arrival times within the same $B_i$ can be seen as uniformly distributed in $(0, t)$ (see, for example Tijms [64]). Thus, we can apply the results of our baseline scenario to each of these instances individually.

Seen from this angle, the original problem can be restated as follows. Given that $N(t)$ clusters have arrived uniformly in $(0, t)$, each containing an expected number $E[Y] = \sum_k k c_k$ of cars, and given that each car is marked and still resident in the AoI with probability $\gamma(t)$, of interest is the expected number $R(t)$ of marked cars in the AoI at time $t$.

**Theorem 1.** $E[R(t)] = \Lambda(t)E[Y]$.

**Proof.** We begin by evaluating the conditional expectation $E[R(t) \mid \{N(t) = n\}]$. Since

$$R(t) = \sum_{i=1}^{N(t)} S_i(t)$$

we have

$$
\begin{aligned}
E[R(t) \mid \{N(t) = n\}] &= E[\sum_{i=1}^{N(t)} S_i(t) \mid \{N(t) = n\}] \\
&= E[\sum_{i=1}^{n} S_i(t)] \\
&= \sum_{i=1}^{n} E[S_i(t)] \\
&= n \sum_k k c_k \frac{1}{t} \int_0^t p(t - u)[1 - F_G(u)] \, \mathrm{d}u.
\end{aligned}
$$

Finally, the Law of Total Expectation allows us to write

$$
\begin{aligned}
E[R(t)] &= \sum_{n \geq 0} E[R(t) \mid \{N(t) = n\}] \Pr[\{N(t) = n\}] \\
&= \lambda \sum_k k c_k \int_0^t p(t - u)[1 - F_G(u)] \, \mathrm{d}u \\
&= \lambda \int_0^t p(t - u)[1 - F_G(u)] \, \mathrm{d}u \sum_k k c_k, \\
&= \Lambda(t)E[Y]
\end{aligned}
$$

completing the proof of the theorem. □

Theorem 1 allows the TMC to form an idea about the expected number of VACCS-enabled cars likely to reside in the AoI. This, in turn, will suggest a feasible number, $W$, of cars that should be drafted into the workforce, as discussed next.

### 6.4. Selecting the Workforce

There are several possible avenues for selecting the participating cars in VACCS. These avenues comprise eligibility criteria to be met as well as strategies for the actual workforce selection. For the latter, we adopt the following approach.

On a common control code (frequency) the $i$-th traffic light along Lexington will advertise a TDMA frame of $n_i$ slots. The nearby vehicles that qualify for participating in the VC bid on various slots in the TDMA frame. We say that a vehicle has *secured* a slot if it is the only vehicle that has transmitted in that slot. At the end of the TDMA frame traffic light $i$ announces the successful bidders. These bidders are, of course, anonymous and also acquire a new *identity*. To understand what is going on, assume that a given vehicle has acquired slot $j$ in the TDMA advertised by traffic light $i$. In VACCS the vehicle is named $(i, j)$. Observe also that, due to traffic moving along, by the time vehicle $(i, j)$ has completed the computation is was tasked to execute, it may well be under the coverage of traffic light $k$ on Lexington. This should be no problem and the vehicle can upload its result with that traffic light.

Further, assume that a some number of participating vehicles have been drafted into the workforce by the $n$ cluster controllers. The task now is to determine which vehicles are qualified to perform the computation. A vehicle is qualified if, with some high probability, it will remain in the congested area for at least $T$ time units, which is the run time of the computation plus the time required to deliver the results to the nearest cluster controller. We note that by the time the computation has completed, the vehicle may have moved out of communications range with cluster controller $i$, which assigned its task, and into the range of some cluster controller $j$. This does not pose a problem as the results from all of the cluster controllers will be sent to the VACCS Manager at the TMC for processing. As long as a vehicle will remain in the AoI, near a VACCS-enabled cluster controller, it can be considered qualified for the task.

The availability of VACCS-enabled vehicles is reported back to the VCS Manager by each cluster controller. If the number of qualified vehicles exceeds the size of the required workforce $W$ determined by the TMC, the request will be accepted and the vehicular crowdsourcing formally set up. On the other hand, if the number of qualified vehicles is not greater than $W$, the VCS Manager negotiates with the TMC, offering alternate tasks that may be feasibly completed by the available qualified workforce. This may involve reducing the time horizon of the simulation, the number of different timing plans simulated, or number of independent replications run.

### 6.5. A Model of Re-Timing of Traffic Lights

Though congestion may begin in a small area, in time it may spill over and affect side streets. Once the signal timings have been changed, the vehicles in the area of highest concentration of congestion will begin to move to other regions of the roadway network. The flow of vehicles leaving the congested area may well generate secondary congestion events over a larger area. If new congestion events result from the original re-timing, the TMC will restart the process of setting up vehicular crowdsourcing local to each congested area. Thus, the wide-area resolution of congestion can be perceived as a *multi-stage process*, with each of the successive stages addressing a less acute instance of congestion than the previous one. Therefore, one severe instance of congestion is reduced, in stages, to several less acute instances, spread over a much larger area. The framework that we have established in VACCS lends itself well to this recursive process.

To get a handle on the problem, we propose to model it, at the *macroscopic* level, as an instance of the classic diffusion problem. With insight gleaned from the macroscopic view,

we will be ready to revert to the *microscopic* problem of re-timing the traffic signals in an AoI, as presented in the working scenario of Section 5.1.

Indeed, referring to Figure 3a imagine that some area $A_1$ is congested. It is reasonable to assume that the large concentration of vehicles in $A_1$ will have to be dissipated over the union $A_1 \cup A_2 \cup A_3$. Once a local rescheduling of the traffic lights at the level of $A_1$ is performed, some of the excess vehicles in $A_1$ will be directed towards area $A_2$ as shown in Figure 3b. Observe that since the area $A_1 \cup A_2$ is, in general, significantly larger than $A_1$, we have reduced the problem of re-timing the traffic lights in $A_1$ to the problem of rescheduling the traffic lights in $A_1 \cup A_2$. Consequently, it is reasonable to expect that while $A_2$ may experience some form of congestion that requires the re-timing of the local traffic lights, the problem is far less acute that the one in $A_1$. By the same token, the excess traffic in $A_1 \cup A_2$ will spread, in time, over the entire area $A_1 \cup A_2 \cup A_3$ as illustrated in Figure 3c.

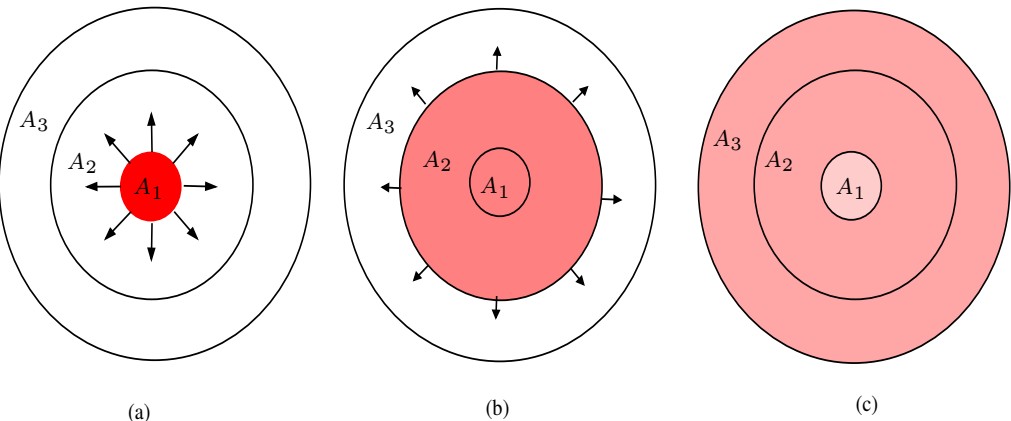

**Figure 3.** A macroscopic view of the congestion dissipation problem.

The benefit of the macroscopic view just outlined is that it helps suggest that the wide-area resolution of congestion can be perceived as a multi-stage process, each of the successive stages addressing a less acute instance of congestion than the previous one. An obvious corollary of the previous discussion is that, by diffusion, one severe instance of congestion is reduced, in stages, to several, less acute instance of congestion, spread over a much larger area. This latter insight has a direct bearing on our strategy for setting up vehicular crowdsourcing to mitigate congestion, and, of course, on the attendant rescheduling of traffic lights.

Returning to the working scenario, we adopt to view that the congestion event on Lexington is in area $A$ (see Section 6.5) and that the wider AoI, over which the municipality intends to re-time the traffic lights, straddles areas $A$ and $B$. Once the TMC has determined to set up a VCS in support of mitigating the congestion at hand, a workforce of participating cars must be elected. As seen above, the flow of vehicles leaving area $A$ may well generate slowdowns or even secondary congestion events over the area $A \cup B$, that contains our AoI.

If the congestion on Lexington has not yet dissipated, a workforce is elected for *each* confirmed instance of a slowdown in AoI (and, more broadly, in $A \cup B$). It is important to observe that, in general, a mere slowdown does not warrant setting up vehicular crowd-sourcing. To help dissipate the resulting slowdown events, the cars stuck in congestion on Lexington will lend a "helping hand". If, on the other hand, several bona-fide congestion events resulted from the original re-timing, then the Smart City will restart the process of setting up an instance of VACCS local to each new congested site.

## 6.6. Computational Approaches for Traffic Optimization

The current state-of-the-practice to optimizing traffic signals involves using macroscopic models like Synchro to optimize signals in an offline manner, using pre-collected traffic data. By contrast, VACCS uses vehicular crowdsourcing to run multiple versions of microscopic traffic simulations using real-time data to produce optimal traffic signal

timing plans. This offers the advantage of explicitly accounting for the stochastic variation in driver behavior, as well as explicitly modeling traffic flow impacts of lane closures and merging behavior. Each participating VACCS-enabled vehicle is assumed to have a locally loaded version of VISSIM, with model networks pre-loaded based on the drivers' known historic travel patterns. Information on boundaries of the traffic network to be simulated, traffic volume, turning movement counts, lane closures, incident conditions, time horizon, pseudo-random number seeds and signal timing parameters would be passed from signal controllers to the vehicle to be run when the vehicular crowdsourcing is activated.

Two basic approaches to optimizing signals are possible. The first one will involve each VACCS-enabled vehicle running simulations with a relatively short time horizon, typically in the three to five minute range. As already mentioned, each vehicle is given a unique set of timing plan parameters and random number seed to run. The results will be reported back to the cluster controller, and the TMC will select the best timing plan based on all reported results. The advantage of this approach is that it uses a short enough time horizon that the timing plans will respond to traffic, and the runtimes for the simulation are also very short (typically 30 s to one minute for a 20 signal network). This would reduce the communication load between vehicles and the cluster controllers since data would only be transferred every three to five minutes. The disadvantage of this approach is that a large number of discrete timing plans would need to be developed in order to produce a comprehensive evaluation of potential alternatives, increasing the pool of vehicles required to do the analysis. Different cycle lengths, phase lengths, and phase orders would need to be explored, creating a potentially large pool of candidate timing plan alternatives. Furthermore, each timing plan would need to be run at least 10 times to estimate mean performance due to stochastic variation in driver behavior.

The second possible approach involves conducting very short time horizon simulations within the vehicular crowdsourcing, on the order of 15 to 30 s. In this case, the number of alternatives would be vastly simplified. It would change to either continuing the current phase for the duration of the simulation, or terminating the current phase and moving to another phase at the intersections (a maximum of 8 different plans). This significantly reduces the number of traffic signal alternatives to be examined, thereby reducing the number of VACCS-enabled vehicles required in the vehicular crowdsourcing. The downside of this approach is that the communications load between the vehicle and the cluster controller is significantly increased, and there is an increased need for timely data transfer.

In both approaches outlined, the vehicle would pass the final measures of effectiveness for each simulation (delays by movement, volume, speed, etc) back to the cluster controller. A performance function will be developed to weigh the different scenarios that have been simulated to select the best ultimate signal timing plan for implementation. A key consideration in this approach is the amount of time available for the vehicular crowdsourcing to run a simulation. Several parameters in VISSIM have a direct relationship with run time. One main parameter is the number of VISSIM time steps per second. For example, a time step of 0.1 s could be increased to 0.5 s. This would significantly reduce run times, but at the cost of a loss of model fidelity and a likely underestimation of congestion. In future work we will investigate how VACCS should tune the simulation parameters dynamically to cope with the vehicles' actual availability and required execution time.

In cases of very large urban networks, the network could be pre-segregated into sub-areas representing common commuter routes. For example, areas would be defined based on the proportion of inbound/outbound traffic, as well by the number of interconnected and parallel routes feeding these common movements. In these cases, optimization would occur on each of these subareas independently.

## 7. Concluding Remarks and Challenges Ahead

VACCS is a novel, transformative, next-generation computing paradigm that is not possible with currently available technology. We trust that our vision paper will motivate further research into the coordination between tomorrow's vehicles and the computational,

sensing, storage, and networking resources that they will possess. Enabling this interaction faces many challenges, including the requirement of wireless communications and the mobility of individual vehicles which affects the dynamics of groups of vehicles. In our case study, we will explore how vehicular crowdsourcing can be used to improve traffic flow through traffic optimization simulations. We anticipate many other applications of this technology that could have a profound and lasting impact on our society. A number of challenges are open and will attract attention as we describe in the next paragraphs.

We expect VACCS to demand significant amounts of resources and to have diverse quality of service requirements. To a great extent, the unpredictable nature of resources available is one of the defining characteristics of vehicular crowdsourcing. We will investigate techniques for discovering VACCS-enabled vehicles near an intersection, predicting which vehicles will be available near the congested intersection for the duration of the computation task, and assigning tasks to vehicles distributed over multiple intersections.

With the US government's Connected Vehicle program, DSRC/WAVE is the *de facto* standard for vehicular communications. Therefore, we discuss the communications required for VACCS in an AoI as a DSRC/WAVE service of the cluster controllers. In this scheme a cluster controller would advertise the CfP on the control channel (CCH) while the actual exchange of VACCS data including optimization results will occur on the service channel (SCH) indicated in the advertisement. The current DSRC/WAVE guidelines call for the CCH interval to be 50 ms long and repeated 10 times a second while the rest is for the SCH. One main challenge with this, as previous research has shown [65], is that when more communication on the CCH is needed for safety applications due to high traffic density, there will necessarily be less time available on the SCH for other communications, such as VACCS. As the congested traffic conditions addressed in this paper necessarily involve high traffic density, we will investigate the feasibility and performance of DSRC/WAVE in comparison to using regular WiFi standards. Another point we will investigate is single hop versus multi-hop for VACCS communications between vehicles and a cluster controller. Given the congested traffic conditions for VACCS operation, it is expected that a cluster controller with a typical wireless coverage range (e.g., 150 m) would cover a large number of participant vehicles within its range. For example, with two single-direction approaches and three lanes in each approach (similar to our working scenario in Figure 1) and average vehicle length is about five meters, the number of vehicles congested at this single intersection would be up to 180 vehicles.

As pointed out by Doan et al. [66], the race is on to build general crowdsourcing platforms in various application domains. This paper could be the base for building other vehicular-crowdsourcing based applications in support of smart services in Smart Cities and the Smart Communities of the future. For example, it appears natural and useful to harness vehicular crowdsourcing in support of planned and unplanned evacuation efforts [67,68].

It would be very important to put to work Big Data analytics using the on-board resources in vehicles. Big Data analytics in conjunction with Machine Learning [69] could enable individual vehicles to process more traffic-related data faster and more accurately. One of the promising directions is the potential for designing and implementing efficient hardware algorithms for basic operations including prefix computation, fast sorting and searching, as well as vision algorithms using various parallel architectures such as, for example, reconfigurable architectures and architectures with various other fast bus systems [70–77]. This promises to be an exciting area for future work.

Finally, two important topics in crowdsourcing, namely security and incentivizing the workforce were not discussed in this paper. Instead, we have tacitly assumed that the drivers stuck in congested traffic have a vested interest to participate in crwodsourcing in order to dissipate congestion as soon as possible and, therefore, no direct incentivization is necessary. Similarly, guarding against simple security threats, such as pranksters injecting wrong information into the system, can be achieved by building some redundancy and fault tolerance into the system. This can be achieved by having several vehicles perform

simulation on the same input data and by eliminating outliers. However, we leave a fuller discussion of security and incentivizing the workforce for future work.

**Funding:** This research was funded by the U.S. National Science Foundation grant CNS-1951789.

**Institutional Review Board Statement:** Not applicable.

**Informed Consent Statement:** Not applicable.

**Data Availability Statement:** This study did not report any data.

**Conflicts of Interest:** The author declares no conflict of interest.

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
