# Peer review of "Vehicular Crowdsourcing for Congestion Support in Smart Cities"

_smartcities, doi:10.3390/smartcities4020034_

Round 1

Reviewer 1 Report

comments per lines in text:

  • 29: signal cycle length (phase length) is based upon the number of vehicles per hour which then results in daily traffic flow variations,
  • 39: FHA - capital letter Administration,
  • 47-54: there is no need for 'supercomputers', but other infrastructure is needed (signal controller communication, real-time traffic data processing, etc.) which is solved with existing commercial traffic control solutions (old SCATS, SCOOT, then UTOPIA),
  • 66-70: the cooperative concept known in Europe is already defined and several projects have been conducted (CVIS, COOPERS) where communication between vehicles and infrastructure is considered (V2I, V2V),
  • 112-113: not only trains, all PT (public transport) vehicles,
  • 138-139: list of acronyms is unnecessary,
  • 189: misspelling - components,

number of references is too high, consider reducing it (some references are unnecessary)

Author Response

Dear Reviewer 1.

Thank you for your comments. I have now had time to revise the manuscript in the light of the comments and suggestions contained in your review report. All the changes are highlighted in blue in the revised manuscript.

As suggested, I have reduced substantially the number of references cited and this has had for effect reducing the length of the manuscript by a whopping three pages!

Thank you, again, for your comments!

Reviewer 2 Report

Very well written paper that study the use of vehicular crowdsourcing for congestion support in smart cities. Here are some points to improve the quality:

1) Using "Manhattan Model" as an example for the working scenario is good, however, it gets more complicated with other models such as AIMSUN Model of Singapore’s Main Road Network.

2) This paper can be a base line for building other applications using crowdsourcing, but would be good to provide some results of how efficient the system work.

3) References are good and more references can be added such as:

Dion, François, Karthik Sivakumaran, and Xuegang Jeff Ban. 
Evaluation of traffic simulation model use in the development of Corridor System Management Plans (CSMPs).
No. UCB-ITS-PRR-2012-2. 2012.

Author Response

Dear Reviewer 2

Thank you for your comments. I am glad you liked the manuscript. I fully agree that this paper could be the starting point of an entire series of subsequent papers that gravitate around the concept of vehicular crowdsourcing.

Please refer to the revised version of the manuscript where changes are highlighted in blue

Thank you!